# GA4GH Phenopackets: A Practical Introduction

*Markus S. Ladewig, Julius O. B. Jacobsen, Alex H. Wagner, Daniel Danis, Baha El Kassaby, Michael Gargano, Tudor Groza, Michael Baudis, Robin Steinhaus, Dominik Seelow, Nikolaos E. Bechrakis, Christopher J. Mungall, Paul N. Schofield, Olivier Elemento, Lindsay Smith, Julie A. McMurry, Monica Munoz-Torres, Melissa A. Haendel, and Peter N. Robinson\**

The Global Alliance for Genomics and Health (GA4GH) is developing a suite of coordinated standards for genomics for healthcare. The Phenopacket is a new GA4GH standard for sharing disease and phenotype information that characterizes an individual person, linking that individual to detailed phenotypic descriptions, genetic information, diagnoses, and treatments. A detailed example is presented that illustrates how to use the schema to represent the clinical course of a patient with retinoblastoma, including demographic information, the clinical diagnosis, phenotypic features and clinical measurements, an examination of the extirpated tumor, therapies, and the results of genomic analysis. The Phenopacket Schema, together with other GA4GH data and technical standards, will enable data exchange and provide a foundation for the computational analysis of disease and phenotype information to improve our ability to diagnose and conduct research on all types of disorders, including cancer and rare diseases.

## 1. Introduction

The Global Alliance for Genomics and Health (GA4GH), a policy-framing and technical standards-setting organization, is developing a suite of coordinated standards to enable responsible genomic and related-health data sharing.[1] The Phenopacket is a new GA4GH schema for sharing disease and phenotype information. A Phenopacket characterizes an individual person or biosample, linking that individual to detailed phenotypic descriptions, genetic information, diagnoses, and treatments. The Phenopacket schema supports the FAIR principles (findable, accessible, interoperable, and reusable), and computability.[2–5] Specifically, Phenopackets are designed to be

M. S. Ladewig
Department of Ophthalmology
Klinikum Saarbrücken
66119 Saarbrücken, Germany

J. O. B. Jacobsen
William Harvey Research Institute
Charterhouse Square
Barts and the London School of Medicine and Dentistry Queen
Queen Mary University of London
London EC1M 6BQ, UK

A. H. Wagner
Departments of Pediatrics and Biomedical Informatics
The Ohio State University College of Medicine
Columbus, OH 43210, USA

A. H. Wagner
The Steve and Cindy Rasmussen Institute for Genomic Medicine
Nationwide Children's Hospital
Columbus, OH 43215, USA

D. Danis, B. El Kassaby, M. Gargano, P. N. Robinson
The Jackson Laboratory for Genomic Medicine
10 Discovery Drive, Farmington, CT 06032, USA
E-mail: peter.robinson@jax.org

T. Groza
European Molecular Biology Laboratory
European Bioinformatics Institute (EMBL-EBI)
Cambridge CB10 1SD, UK

M. Baudis
Department of Molecular Life Sciences and Swiss Institute of Bioinformatics
University of Zurich
Zurich, Switzerland

R. Steinhaus, D. Seelow
Exploratory Diagnostic Sciences
Berlin Institute of Health at Charité – Universitätsmedizin Berlin
10178 Berlin, Germany

R. Steinhaus, D. Seelow
Institute of Medical Genetics and Human Genetics
Charité – Universitätsmedizin Berlin
Corporate Member of Freie Universität Berlin and Humboldt-Universität zu Berlin
13353 Berlin, Germany

N. E. Bechrakis
Department of Ophthalmology
University Clinic Essen
45147 Essen, Germany

both human and machine-interpretable, enabling computing operations and validation on the basis of defined relationships between diagnoses, lab measurements, and genotypic information.[2]

The Phenopacket schema enables comparison of sets of phenotypic attributes from individual patients. Such comparisons can aid in diagnosis and facilitate patient classification and stratification for identifying new diseases and treatments.[2] The Phenopacket schema is designed to support interoperability between people, organizations, and systems that comprise the worldwide effort to understand human disease. The structure of the information in a phenopacket was designed for integration within clinical laboratories, journals, data repositories, patient registries, electronic health records (EHRs), and knowledge bases. Increasing the volume of computable data across a diversity of systems will support global disease analysis by integrating genotype, phenotype, and other multi-modal data for precision health applications.

Because of the broad range of intended use cases and the large number of terminologies and ontologies in use by different communities, the Phenopacket Schema is intentionally flexible with respect to which elements are required and which terminologies or ontologies must be used. Nonetheless, a given hospital, project, or research consortium may wish to apply different constraints. For instance, a Mendelian genetics consortium might stipulate the use of Human Phenotype Ontology (HPO) terms[3] to describe phenotypic abnormalities, and a cancer genomics consortium might require that each phenopacket have a biosample for a tumor biopsy in which National Cancer Institute

C. J. Mungall
Lawrence Berkeley National Laboratory
Environmental Genomics and Systems Biology
Berkeley, CA 94720, USA

P. N. Schofield
Department of Physiology Development and Neuroscience
University of Cambridge
Downing Street, Cambridge CB2 3EG, UK

P. N. Schofield
The Jackson Laboratory
Bar Harbor, ME 04609, USA

O. Elemento
Caryl and Israel Englander Institute for Precision Medicine
Weill Cornell Medicine
New York, NY 10021, USA

L. Smith
Ontario Institute for Cancer Research
Adaptive Oncology
Toronto, CA M5G0A3, USA

L. Smith
Global Alliance for Genomics and Health
Toronto, CA M5G0A3, USA

J. A. McMurry, M. Munoz-Torres, M. A. Haendel
Center for Health AI
University of Colorado Anschutz Medical Campus
Aurora, CO 80045, USA

P. N. Robinson
Institute for Systems Genomics
University of Connecticut
Farmington, CT 06032, USA

Thesaurus (NCIT) terms[4] are used to describe the histological features and other phenotypic characteristics.

In this article, we demonstrate the features of the GA4GH Phenopacket based on a typical clinical case report of a child with retinoblastoma.[5] Retinoblastoma is a malignant tumor of the developing retina that typically occurs in children. Retinoblastoma is the most common eye cancer in childhood, but still a relatively rare disease, occurring in approximately one in 16 000–18 000 live births worldwide.[6] The most common presenting signs are white pupillary reflex (leukocoria) and squinting (strabismus). Most affected children are diagnosed before the age of 5 years. Standard examination methods include, in particular, ophthalmoscopy, sonography, and—especially for staging purposes—magnetic resonance imaging. The chance of recovery and of preservation of eyesight depends very much on the spread of the tumor inside and outside the eye. The main therapeutic goal is always to save the life of the patient, which means primarily a complete removal of the tumor, while clinical management aims to preserve the eyeball and as much of the vision as possible.[7] Retinoblastoma can occur sporadically or as a Mendelian trait. According to the Knudson hypothesis,[8] development of retinoblastoma is initiated if both *RB1* alleles have acquired disease-causing mutations. In sporadic forms, both mutations occur in somatic cells; in heritable retinoblastoma, the first *RB1* mutation is inherited and only the second mutation occurs as a somatic event. In some cases, the first mutation occurs de novo either as a germline variant or a somatic mosaic.[9] The *RB1* gene is located at chromosome 13q14. Some patients have large 13q deletions that contain the *RB1* gene.[10] Germline or de novo mosaic 13q deletions can involve the surrounding genes delineating a contiguous gene syndrome characterized by retinoblastoma, developmental anomalies, and facial dysmorphisms.[11] Mosaicism results from postzygotic mutation that occurs during early embryonic development and can lead to germline or somatic mosaicism, potentially causing a less severe and/or variable phenotype compared with the equivalent constitutive mutation.[12]

## 2. Basic Introduction to Protobuf

The Phenopacket Schema is a data model that is represented using the open source protocol buffers framework developed by Google. Protocol buffers ("protobuf" for short) provide a language-neutral, platform-neutral, and extensible mechanism for serializing structured data. They are used extensively in inter-server communications as well as for archival storage of data on disk.[13] The data schema is described in .proto files and specifies how data is to be serialized (stored) as a hierarchical structure with key-value pairs similar to JSON or XML. However, in contrast to these formats which are written in plain text, data that is encoded using protobuf can be exchanged in a memory-efficient binary format. The protobuf file can be converted by the protobuf compiler into many different computer languages, including Java and Python, which provides a widely used framework for working with phenopackets. Additionally, protobuf can easily be converted into JSON or YAML for human consumption.[14]

In protobuf terminology, a message is a data structure that can contain multiple fields. Messages can be combined hierarchically so that one message contains another message. The .proto files define the structure of messages and the data types of the fields,

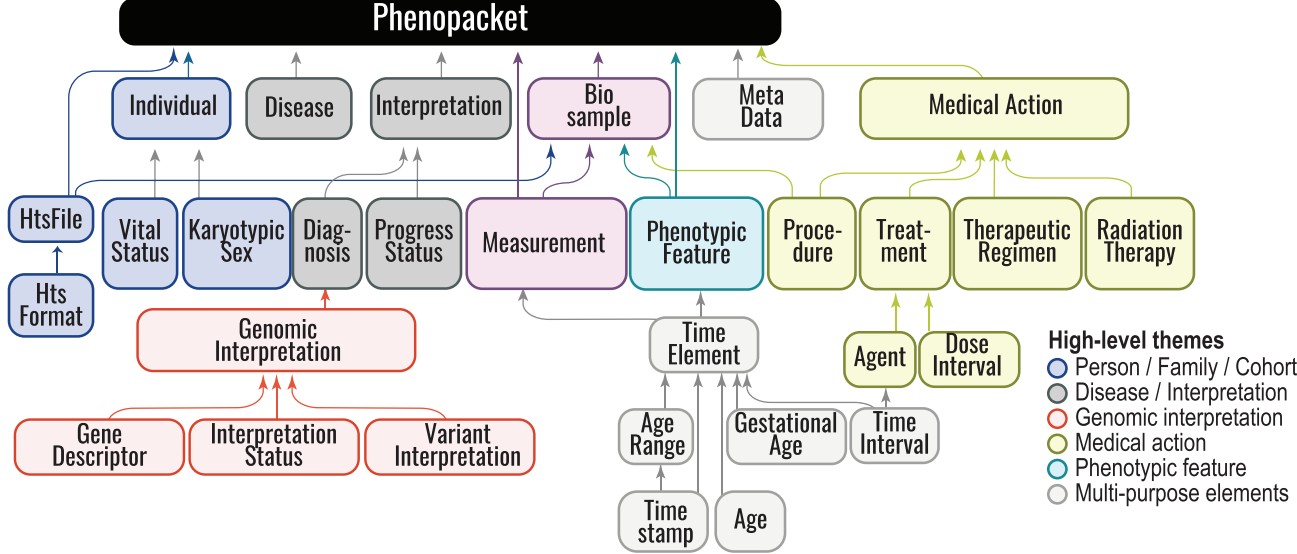

**Figure 1.** Phenopackets Protobuf message comparison. (a) Definition of OntologyClass, a data type in the Phenopacket Schema, in protobuf. (b) Representation of an instance of OntologyClass (representing the HPO term for neutropenia) in a YAML format. (c) Equivalent representation in JSON format.

**Figure 2.** Phenopacket Schema overview. The GA4GH Phenopacket Schema is a hierarchical structure that consists of two required fields, id and Meta-Data, as well as eight optional fields, Individual, Disease, Interpretation, Biosample, PhenotypicFeature, Measurement, MedicalAction, and files, each of which is discussed in this article. A detailed version of the schema, including elements from VRS/VRSATILE, is shown in ref. [2].

including integers, strings, floating point numbers, and timestamps (**Figure 1**).

In this article, we present a detailed example that shows how to encode various kinds of clinical data using the Phenopacket Schema. We will not discuss computational tools for creating phenopackets, which include Java and Python frameworks available from the protobuf framework itself. For readability, we will display phenopacket data using YAML, an intentionally human-friendly data serialization language that is one of the many different formats in which Phenopackets can be represented including YAML, JSON, binary (protobuf), RDF, and SQL. YAML and JSON versions of the complete phenopacket presented here are available as Supplementary files 1 and 2.

## 3. Overall Structure of the Phenopacket

The GA4GH Phenopacket Schema contains several optional elements to represent a case report. The current example uses many of the optional fields. The Phenopacket represents the entire case report with information about the patient (Individual), the clinical diagnosis (Disease), several phenotypic features (PhenotypicFeature), clinical measurements (Measurement), an examina-

tion of the extirpated tumor (Biosample), therapies (MedicalAction), and the results of genomic analysis (GenomicInterpretation) (**Figure 2**).

Phenopacket Schema is capitalized as a proper noun to refer to the data schema, however one should write "phenopacket" in lower case when one is referring to a phenopacket that represents an individual case. The following examples illustrate common use cases but do not present all of the fields in the Phenopacket Schema. Comprehensive documentation is available online.[15] This article does not cover how to create phenopackets computationally, although we touch on this topic in the Discussion. Instead, we present a detailed example that is intended to help implementers and users of phenopackets understand their scope and capabilities.

## 4. Retinoblastoma and Mosaic 13q Deletion: A Case Report

For this example, we present a 6-month-old child with retinoblastoma owing to a *de novo* mosaic deletion on 13q involving multiple genes, including *RB1*. She first came to medical attention because of leukocoria, strabismus, and retinal detachment in her

**Figure 3.** Phenopacket messages. This Figure as well as Figures 4–12 show excerpts of the phenopacket with line numbers added. (a) Line 1 shows the beginning of the phenopacket (in a YAML file that only contains a single phenopacket, line 1 would contain a "–" instead of "phenopacket"). Line 2 contains the identifier of the phenopacket, which is required to be present but whose syntax is arbitrary and generally should be specified by the application. Lines 3–9 show the Individual message. (b) An example vitalStatus message for another example patient (not related to the example) that indicates the time and cause of death as well as the survival time following the primary diagnosis.

left eye. Diagnostic workup included whole-genome sequencing (WGS) of germline DNA, which led to the identification of the de novo 13q deletion. Treatment included first intra-arterial melphalan infusions, which had to be discontinued because of adverse effects (vasospasm). Treatment with a regimen of three chemotherapeutic agents was attempted, but removal of the affected eye (enucleation) became necessary because of lack of response to the regimen. Histological investigation of tissue from the enucleated eye revealed several typical findings including apoptotic and necrotic cells and Flexner–Wintersteiner rosettes. WGS of DNA from the tumor sample revealed a somatic single-nucleotide variant in *RB1* ("second hit").[9] For demonstration purposes, some clinical features were added to the original case report. The following sections show how individual components (messages) of the Phenopacket Schema can be used to represent the clinical information from the case report.

## 5. Individual

Each phenopacket describes one individual. In many cases, the individual is a patient or a proband of a study, but phenopackets can be used to represent information about controls as well. The subject of the phenopacket is represented by an Individual message that contains fields for an identifier, the date of birth or age at the latest clinical encounter, the vital status, the sex, gender, and karyotypic sex. The majority of fields in Individual (and of the other elements of the Phenopacket Schema) are optional and can be left out if no information is available or if the field is not relevant for the intended use case. For instance, the date of birth field should not be used if a phenopacket is intended to be shared, and the karyotypic sex element should only be used if the results of chromosome analysis are available. The age (or age range) of the individual can be represented in a number of ways, some of which preserve privacy and are intended for responsible data sharing, and some of which are more precise. If the phenopacket is not intended to be shared broadly and the birthdate and the date of the last encounter need to be recorded, a TimeStamp message can be used to represent this information precisely. In many cases, the birthdate will be omitted and the age can be represented using an ISO 8601 string. For instance,

the age of 42 years, 7 months, and 13 days would be represented as P42Y7M13D. One can also represent the age as P42Y without specifying the month or days, or one can use an AgeRange message to indicate that the individual's age lies within a given range, which may be desirable to help preserve privacy. All information in a phenopacket refers to this individual.

In the current case, the following information was provided about the patient in the original publication.

"A 6-month-old girl conceived by in vitro fertilization (IVF) (own oocytes and anonymous donor sperm) was admitted to the hospital because of leukocoria and strabismus. Past medical history and physical examination were unremarkable except for clinodactyly of the right fifth finger. Indirect ophthalmoscopic examination and examination under anesthesia was performed by ophthalmologists. Orbital ultrasound and magnetic resonance imaging (MRI) scans showed a 14 × 13 × 11 mm left eye tumor located in the lower-external retinal side. Retinal detachment was also detected."

The Phenopacket representation of the clinical information for this case is shown in **Figures 3**a–**12**. The patient id is arbitrarily defined as "proband A". Based on the case report, several key features were included in the Phenopacket representation. We define the age of the patient as 6 months (P6M) and the sex as female. For the sake of example, we have indicated the karyotypic (chromosomal) sex of the proband, although chromosome analysis would not typically be performed for this clinical indication (Figure 3a). Additionally, the gender field can be used to specify the self-identified gender of an individual using an ontology term,[16] and the vitalStatus message can be used to indicate if a patient is alive or deceased (an example unrelated to the retinoblastoma case is shown in Figure 3b).

## 6. PhenotypicFeature

The PhenotypicFeature is the central element of the Phenopacket Schema. A PhenotypicFeature can be used to describe each phenotypic feature (often, but not necessarily, clinical abnormalities) including signs and symptoms, laboratory findings, imaging, and electrophysiological results, along with modifier and qualifier concepts. Each phenotypic feature is described using an

```
10  phenotypicFeatures:          29  - type:
11  - type:                      30      id: "HP:0000486"
12      id: "HP:0030084"         31      label: "Strabismus"
13      label: "Clinodactyly"    32    modifiers:
14    modifiers:                 33    - id: "HP:0012835"
15    - id: "HP:0012834"         34      label: "Left"
16      label: "Right"           35    onset:
17    onset:                     36      age:
18      age:                     37        iso8601duration: "P5M15D"
19        iso8601duration: "P3M" 38  - type:
20  - type:                      39      id: "HP:0000541"
21      id: "HP:0000555"         40      label: "Retinal detachment"
22      label: "Leukocoria"      41    modifiers:
23    modifiers:                 42    - id: "HP:0012835"
24    - id: "HP:0012835"         43      label: "Left"
25      label: "Left"            44    onset:
26    onset:                     45      age:
27      age:                     46        iso8601duration: "P6M"
28        iso8601duration: "P4M"
```

**Figure 4.** List of PhenotypicFeatures. Clinodactyly, which is not known to be related to retinoblastoma and is presumably an incidental finding, was noted at the age of 3 months (P3M). Leukocoria was noted at the age of 4 months, strabismus at the age of 5 months and 15 days, and retinal detachment at the age of 6 months.

ontology term. One can indicate whether a certain abnormality was excluded during the diagnostic process (e.g., whether a morphological cardiac defect was excluded by echocardiography), or optionally use other HPO terms to denote the severity of the phenotypic feature or add other modifiers that describe the frequency (e.g., number of occurrences of seizures per week), laterality (e.g., unilateral), or another pattern of a certain phenotypic feature in the patient being described. Finally, the onset (and if applicable the resolution) of specific features can be indicated (**Figure 4**).

## 7. Measurement

The Measurement message is used to capture quantitative, ordinal (e.g., absent/present), or categorical measurements. For some applications such as phenotype-driven genomic diagnostics of rare disease, qualitative representations of phenotypic abnormalities are appropriate, e.g., "Ocular hypertension" (HP:0007906), which denotes an increase of the intraocular pressure that is 2 standard deviations or more above the population mean. For other use cases, such as following the development of some parameter over time, the original quantitative values may be preferable, and can be represented using the Measurement element. Eyes that harbor a retinoblastoma characteristically display increased intraocular pressure.[17] In our example, the patient is noted to have an intraocular pressure of 25 mm Hg, which is above the normal range, observed in the eye affected by the retinoblastoma (left eye). Measurement messages can be used to represent normal measurements; for instance, in our example the right eye was not affected by retinoblastoma and displayed a normal intraocular pressure of 15 mm Hg, measured with the Perkins tonometer.[17] Both measurements include the reference (normal) range of 10–21 mm Hg (**Figure 5**).

```
47  measurements:
48  - assay:
49      id: "LOINC:79893-4"
50      label: "Left eye Intraocular pressure"
51    value:
52      quantity:
53        unit:
54          id: "UO:0000272"
55          label: "millimetres of mercury"
56        value: 25.0
57        referenceRange:
58          unit:
59            id: "56844-4"
60            label: "Intraocular pressure of Eye"
61          low: 10.0
62          high: 21.0
63    timeObserved:
64      age:
65        iso8601duration: "P6M"
66  - assay:
67      id: "LOINC:79892-6"
68      label: "Right eye Intraocular pressure"
69    value:
70      quantity:
71        unit:
72          id: "UO:0000272"
73          label: "millimetres of mercury"
74        value: 15.0
75        referenceRange:
76          unit:
77            id: "56844-4"
78            label: "Intraocular pressure of Eye"
79          low: 10.0
80          high: 21.0
81    timeObserved:
82      age:
83        iso8601duration: "P6M"
```

**Figure 5.** Measurement. A) Measurements of intraocular pressure (IOP) in the left eye (lines 48–65) and right eye (lines 66–83).

## 8. Biosample

A Biosample contains information about the examination of biological specimens from which the substrate molecules (e.g., genomic DNA, RNA, proteins) for molecular analyses (e.g., sequencing, array hybridisation, mass-spectrometry) are extracted. Examples would be a tissue biopsy, a single cell from a culture for single cell genome sequencing, or a protein fraction from a gradient centrifugation. Several instances (e.g., technical replicates) or types of experiments (e.g., genomic array as well as RNA-seq experiments) may refer to the same Biosample.

In our example, the Biosample message contains information about the histology of a "retinoblastoma" (NCIT:C39853) removed by "enucleation" (NCIT:C48601) of the "left eye" (UBERON:0004548) and classified as "Retinoblastoma pT3 TNM Finding v8" (NCIT:C140720). Histology further showed a "Flexner–Wintersteiner rosette formation" typical of retinoblastoma, in addition to "Apoptosis and Necrosis" (**Figure 6**).

While a Biosample message could be used to describe routine blood or buccal swab samples used for analysis of germline DNA, if the specifics of the collected tissue are not deemed relevant for the analysis, the information can be omitted (in this case, the subjectOrBiosampleId of the relevant GenomicInterpretation element should be set to the proband id; see below).

## 9. Interpretation

A Phenopacket can contain one or more Interpretation elements that specify interpretations of genomic findings. For instance, a report from a diagnostic laboratory about a variant interpreted to be causal for a certain Mendelian disease may be represented as an Interpretation element. Another example would be a report of an actionable somatic variant for which a targeted cancer therapy is available.

As a GA4GH standard, the Phenopacket Schema integrates with and leverages other GA4GH standards when applicable. The Phenopacket Schema uses the computational precision of the GA4GH Variation Representation Specification (VRS) while maintaining the flexibility of describing variation using human-readable variant description formats such as SPDI (NCBI sequence, position, deletion and insertion model)[18] and HGVS[19] through collaborative development and adoption of the VRS Added Tools for Interoperable Loquacious Exchange (VRSATILE).[20] VRSATILE provides two primary object classes that are used in the Phenopacket Schema: the VariationDescriptor and the GeneDescriptor. These descriptor classes allow the extension of computationally precise concepts (e.g., VRS alleles, HGNC gene identifiers) with common additional attributes for systems to describe these concepts (e.g., identifier cross-references, HGVS descriptions, gene symbols, and informative contexts such as variant zygosity). This collaborative framework bridges existing variant representation formats and implementations to the more computationally precise concepts enabled by VRS.

In our example, the patient was found to have one mosaic 13q germline deletion and one somatic missense mutation. A phenopacket can contain one or more Interpretation messages, whereby each Interpretation refers to a single disease that is specified in the diagnosis field. The Interpretation contains a list of

```
84  biosamples:
85  - id: "biosample.1"
86    sampledTissue:
87      id: "UBERON:0000970"
88      label: "eye"
89    phenotypicFeatures:
90    - type:
91        id: "NCIT:C35941"
92        label: "Flexner-Wintersteiner Rosette Formation"
93    - type:
94        id: "NCIT:C132485"
95        label: "Apoptosis and Necrosis"
96    measurements:
97    - assay:
98        id: "LOINC:33728-7"
99        label: "Size.maximum dimension in Tumor"
100     value:
101       quantity:
102         unit:
103           id: "UO:0000016"
104           label: "millimeter"
105         value: 15.0
106     timeObserved:
107       age:
108         iso8601duration: "P8M2W"
109   tumorProgression:
110     id: "NCIT:C8509"
111     label: "Primary Neoplasm"
112   pathologicalTnmFinding:
113   - id: "NCIT:C140720"
114     label: "Retinoblastoma pT3 TNM Finding v8"
115   - id: "NCIT:C140711"
116     label: "Retinoblastoma pN0 TNM Finding v8"
117   procedure:
118     code:
119       id: "NCIT:C48601"
120       label: "Enucleation"
121     bodySite:
122       id: "UBERON:0004548"
123       label: "left eye"
124     performed:
125       age:
126         iso8601duration: "P8M2W"
127   files:
128   - uri: "file://data/fileSomaticWgs.vcf.gz"
129     individualToFileIdentifiers:
130       biosample.1: "specimen.1"
131     fileAttributes:
132       genomeAssembly: "GRCh38"
133       fileFormat: "VCF"
```

**Figure 6.** Biosample. The id (line 85) is required and can be used to relate genomic interpretations to the biosample that corresponds to the interpretation (see Figure 8). Lines 86–88 represent the tissue of origin of the specimen, lines 89–95 represent phenotypic features of the specimen, and lines 96–108 represent a measurement taken of the maximal size of the tumor. Note that the same PhenotypicFeature and Measurement message definitions are used here as described above. The tumor progression field (lines 109–111) is used to specify whether a tumor is primary, metastatic, or recurrent. Lines 112–126 contain the pathological TNM (primary Tumor, lymph Nodes, distance Metastasis) assessment. Finally, lines 127–133 specify a File with results of whole genome sequencing performed on this tissue sample (See the section on File messages, below, for explanations). The interpretation based on this sequencing is presented in Figure 8.

one or multiple GenomicInterpretation messages that contain information about genetic findings that support the diagnosis. In our case, the diagnosis is retinoblastoma, which is indicated by the corresponding NCIT ontology term. Each GenomicInterpretation has a mandatory subjectOrBiosampleId field that specifies either the id of the patient (which must be the same

**2200016 (6 of 12)**

```
134  interpretations:
135  - id: "interpretation.id"
136    progressStatus: "SOLVED"
137    diagnosis:
138      disease:
139        id: "NCIT:C7541"
140        label: "Retinoblastoma"
141      genomicInterpretations:
142      - subjectOrBiosampleId: "proband A"
143        interpretationStatus: "CAUSATIVE"
144        variantInterpretation:
145          acmgPathogenicityClassification: "PATHOGENIC"
146          therapeuticActionability: "ACTIONABLE"
147          variationDescriptor:
148            variation:
149              copyNumber:
150                allele:
151                  sequenceLocation:
152                    sequenceId: "refseq:NC_000013.14"
153                    sequenceInterval:
154                      startNumber:
155                        value: "25981249"
156                      endNumber:
157                        value: "61706822"
158                number:
159                  value: "1"
160            extensions:
161            - name: "mosaicism"
162              value: "40.0%"
```

**Figure 7.** Interpretation and GenomicInterpretation (1). The first portion of the Interpretation message is shown, with the status of the interpretation (solved) on line 136, the diagnosis to which the interpretation refers on lines 137–140, and the list of GenomicInterpretations starting on line 141. The first of two GenomicInterpretations is shown in this Figure, describing the mosaic deletion on chromosome 13. The deletion is specified as a CopyNumber message, which indicates the chromosome (the corresponding NCBI RefSeq NC_000013.14 identifier is listed), start and end positions, and the copy number (1, corresponding to a loss of one of the two copies of RB1). The extensions field is used to specify additional information about variants, such as the degree of mosaicism (here) or the allele frequency in the second GenomicInterpretation.

```
163    - subjectOrBiosampleId: "biosample.1"
164      interpretationStatus: "CAUSATIVE"
165      variantInterpretation:
166        acmgPathogenicityClassification: "PATHOGENIC"
167        therapeuticActionability: "ACTIONABLE"
168        variationDescriptor:
169          id: "rs121913300"
170          variation:
171            allele:
172              sequenceLocation:
173                sequenceId: "refseq:NC_000013.11"
174                sequenceInterval:
175                  startNumber:
176                    value: "48367511"
177                  endNumber:
178                    value: "48367512"
179                literalSequenceExpression:
180                  sequence: "T"
181          label: "RB1 c.958C>T (p.Arg320Ter)"
182          geneContext:
183            valueId: "HGNC:9884"
184            symbol: "RB1"
185          expressions:
186          - syntax: "hgvs.c"
187            value: "NM_000321.2:c.958C>T"
188          - syntax: "transcript_reference"
189            value: "NM_000321.2"
190          vcfRecord:
191            genomeAssembly: "GRCh38"
192            chrom: "NC_000013.11"
193            pos: "48367512"
194            ref: "C"
195            alt: "T"
196          extensions:
197          - name: "allele-frequency"
198            value: "25.0%"
199          moleculeContext: "genomic"
200          allelicState:
201            id: "GENO:0000135"
202            label: "heterozygous"
```

**Figure 8.** GenomicInterpretation (2). The GenomicInterpretation message uses the subjectOrBiosample field to indicate the source of the sequenced material. In this case, the source was the tumor specimen described in the Biosample message of Figure 6. The variant, whose id is indicated as the corresponding dbSNP accession number (rs121913300) is classified as pathogenic using ACMG criteria (ClinVar VCV000126824.9). Information about the genomic location of the variant is provided as an Allele message in lines 171–180. The corresponding HGVS expression is provided in the label field (line 181; in general, the label can contain arbitrary text). The GeneContext field indicates the affected gene with its Human Gene Nomenclature Committee identifier and gene symbol in lines 182–184. The corresponding variant call format (VCF) representation of the variant is shown in the VcfRecord message in lines 190–195. The zygosity of the variant is specified in the allelicState field.

as the id used in the subject field) or the identifier of the Biospecimen (which must be identical with the id used in one of the Biosample messages of the same phenopacket). For simplicity, we present this finding of a mosaic 13q in our example case as the result of whole-genome sequencing with estimation of mosaicism, but other technologies such as DNA microarrays might be more likely to be used in current clinical settings (**Figure 7**).

In our example, next-generation sequencing of the retinoblastoma tumor sample was performed with a gene panel to search for somatic variants.

"Looking for second hit mutations in *RB1*, we applied a custom designed NGS panel (Onconano V2) that included the *RB1*, *BCOR* and *CREBPP* genes (among other 400 commonly mutated genes in pediatric cancer). The study detected only one pathogenic single-nucleotide variant, RB1 c.958C > T (p.Arg320Ter) (NM_000321.2 chromosomal position 13–48,941,648-C-T; allele frequency of 25%)."

This variant is represented as a second GenomicInterpretation message (**Figure 8**).

## 10. Disease

The Disease message refers to the clinical diagnosis of one or more diseases that the patient has. In our example, the diagnosis of grade E retinoblastoma was made based on the International Classification for Intraocular Retinoblastoma. The Disease message allows the reporting of one or more clinical TNM stages to indicate the extent of the primary tumor as well as potential nodal involvement (i.e., spread of malignant cells to regional or distal lymph nodes) and the detection or absence of distant metastases. In our example, a Retinoblastoma-specific term from the NCI Thesaurus is used to indicate that no evidence of metastasis was found (**Figure 9**; nodal involvement is not a standard part of Retinoblastoma staging).

Clinical staging is based on information available prior to the initial definitive treatment. Pathologic staging includes clinical information and information obtained from pathologic examination of resected primary and, if applicable, regional lymph nodes. The Phenopacket Schema includes a field for clinical TNM codes in the Disease message, and an additional field for pathological TNM codes in the Biosample message (Figure 6).

```
203  diseases:
204  - term:
205      id: "NCIT:C7541"
206      label: "Retinoblastoma"
207    onset:
208      age:
209        iso8601duration: "P4M"
210    diseaseStage:
211    - id: "LOINC:LA24739-7"
212      label: "Group E"
213    clinicalTnmFinding:
214    - id: "NCIT:C140678"
215      label: "Retinoblastoma cM0 TNM Finding v8"
216    primarySite:
217      id: "UBERON:0004548"
218      label: "left eye"
```

**Figure 9.** Disease message. Disease message describing the diagnosis, the stage,[21] age of onset, and primary site of retinoblastoma in the patient.

In cases where there is no disease diagnosis, the Disease message can be omitted. Alternatively, a general diagnosis can be indicated by choosing an ontology term that denotes a group of specific diseases. For instance, the initial clinical diagnosis of ciliopathy (MONDO:0005308) might be made in a child with clinical manifestations such as situs inversus and postaxial polydactyly. The genetic workup (which would be represented in the GenomicInterpretation message; see below) could describe the molecular diagnosis of Bardet–Biedl syndrome 17 (MONDO:0014445).

## 11. MedicalAction

For cancer, infectious disease, rare disease, and many categories of common disease, precision-medicine approaches to treatment based on classification by genetic variants are rapidly gaining in importance. Consequently, clinical decision-making needs to integrate genomic research findings with structured information about treatments. The GA4GH Phenopacket Schema includes a hierarchical representation of medical actions including medications, procedures, and other actions taken for clinical management. The Treatment element represents administration of a pharmaceutical agent, broadly defined as prescription and over-the-counter medicines, vaccines, and other therapeutic agents such as monoclonal antibodies or CAR T-cell-therapy. In our example, the following treatments were described.

"The patient received intraarterial melphalan but due to a local vasospasm in her left leg, the treatment was discontinued. Afterwards, four courses of conventional chemotherapy were administered (vincristine, carboplatin and etoposide). A partial response was achieved, but, despite chemotherapy, the disease progressed few weeks later and the affected eye was enucleated."

The MedicalAction message describes actions taken for clinical management. Each message refers to a specific type of medical action using a Procedure, Treatment, RadiationTherapy, or TherapeuticRegimen message. Our example contains a Treatment message referring to the administration of melphalan (a

chemotherapeutic drug), a TherapeuticRegimen message referring to the administration of a regimen of chemotherapeutics, and a Procedure message referring to enucleation, that is, the surgical removal of the affected eye (**Figure 10**).

### 11.1. Files

The files field of the Phenopacket Schema contains a list of File messages, each of which specifies a file with the results of genomic sequencing or another investigation. In our example, there is a single file and its location on the hard disk is given under uri (Uniform Resource Identifier) (**Figure 11**).

## 12. MetaData

The MetaData message is required to provide details about all of the ontologies and external references used in the Phenopacket. The resources field contains a list of Resource messages with one resource for each ontology or terminology and the corresponding version used to create the phenopacket (**Figure 12**).

## 13. Pedigree

This element is used to represent a pedigree to describe the family relationships of each sample along with their gender and phenotype (affected status). The information in this element can be used by programs for analysis of a multi-sample VCF file with exome or genome sequences of members of a family, some of whom are affected by a Mendelian disease. The Phenopacket Schema has implemented a PED-compatible data model to promote interoperability between existing PED files and PED software, but does not actually store a PED file.

In rare disease diagnostics and research, it is common to perform whole-exome or genome sequencing on multiple family members including persons affected and unaffected by a disease in order to be able to apply cosegregation analysis to filter or prioritize the variants. The GA4GH Phenopacket Schema provides a *Family* message that can include one or more phenopacket messages as well as a representation of the corresponding PED file. Future versions will be integrated with the GA4GH Pedigree standard.[1] An example Family message is shown in Figure S3, Supporting Information. An analogous Cohort message is provided to represent a group of individuals related in some phenotypic or genotypic aspect.

## 14. Discussion

The VCF standard for storing genotyping data allowed a wide range of research groups to write software for analyzing such data.[22] The GA4GH Phenopacket Schema aspires to be similarly transformative in the landscape of disease analysis using an aggregation of phenotype data with core clinical and histopathological information in conjunction with results from genomic data. The Phenopacket Schema was designed to support a number of use cases, including rare-disease diagnostics, supporting searches for samples relevant to a specific disease or condition in biobanks and databases, representation of longitudinal

```
219  medicalActions:
220  - treatment:
221      agent:
222        id: "DrugCentral:1678"
223        label: "melphalan"
224      routeOfAdministration:
225        id: "NCIT:C38222"
226        label: "Intraarterial Route of Administration"
227      doseIntervals:
228      - quantity:
229          unit:
230            id: "UO:0000308"
231            label: "milligram per kilogram"
232          value: 0.4
233        scheduleFrequency:
234          id: "NCIT:C64576"
235          label: "Once"
236        interval:
237          start: "2020-09-02T00:00:00Z"
238          end: "2020-09-02T00:00:00Z"
239      treatmentTarget:
240        id: "NCIT:C7541"
241        label: "Retinoblastoma"
242      treatmentIntent:
243        id: "NCIT:C62220"
244        label: "Cure"
245      adverseEvents:
246      - id: "HP:0025637"
247        label: "Vasospasm"
248      treatmentTerminationReason:
249        id: "NCIT:C41331"
250        label: "Adverse Event"
251  - therapeuticRegimen:
252      ontologyClass:
253        id: "NCIT:C10894"
254        label: "Carboplatin/Etoposide/Vincristine"
255      startTime:
256        age:
257          iso8601duration: "P7M"
258      endTime:
259        age:
260          iso8601duration: "P8M"
261      regimenStatus: "COMPLETED"
262      treatmentTarget:
263        id: "NCIT:C7541"
264        label: "Retinoblastoma"
265      treatmentIntent:
266        id: "NCIT:C62220"
267        label: "Cure"
268  - procedure:
269      code:
270        id: "NCIT:C48601"
271        label: "Enucleation"
272      bodySite:
273        id: "UBERON:0004548"
274        label: "left eye"
275      performed:
276        age:
277          iso8601duration: "P8M2W"
278      treatmentTarget:
279        id: "NCIT:C7541"
280        label: "Retinoblastoma"
281      treatmentIntent:
282        id: "NCIT:C62220"
283        label: "Cure"
```

**Figure 10.** A list of three MedicalAction messages. The Treatment message (lines 220–250) refers to the intraarterial administration of melphalan. A single dose was administered on the indicated date to treat retinoblastoma with curative intent. Vasospasm occurred as an adverse drug effect which necessitated the termination of this treatment. The TherapeuticRegimen message (lines 251–267) refers to the administration of three chemotherapeutic drugs (carboplatin, etoposide, vincristine) according to standard protocols from the age of 7–8 months. The Procedure message (lines 268–283) describes the surgical removal of the affected eye.

```
284  files:
285  - uri: "file://data/germlineWgs.vcf.gz"
286    individualToFileIdentifiers:
287      proband A: "sample1"
288    fileAttributes:
289      genomeAssembly: "GRCh38"
290      fileFormat: "VCF"
```

**Figure 11.** FIle message. The URI field contains Uniform Resource Identifier, that is, a string for locating a file on the internet or other network or a computer file system. The individualToFileIdentifiers field is a map from the identifier used in the phenopacket to those used in the VCF or another file. The fileAttributes field is a list of key value pairs used to specify the genome assembly and the file format.

data in registries, providing a computational representation of published case reports, and other applications. This article provides an in-depth introduction to the representation of clinical data using the GA4GH Phenopacket Schema to annotate a patient with retinoblastoma. We focused on explanations of how

the Phenopacket Schema represents clinical data. In practice, phenopackets will be constructed computationally. The protobuf framework automatically generates code for creating phenopackets in major programming languages including Java, Python, and C++.[13] We show examples of how this can be done in Java and C++ in the main phenopacket schema repository[23] and are preparing a Java library for validation and convenient construction of phenopackets that we will present separately.[24] Additionally, the formats and standards developed for phenopackets are increasingly being adopted by other data discovery and exchange protocols such as the GA4GH Beacon API,[25] as well as several tools and databases that consume or output phenopackets,[26–31] thereby promoting a wider penetration of phenopackets standards and practices throughout biomedical research.

The Phenopacket Schema is flexible and adaptable to local and domain-specific needs. Importantly, and as indicated in the example, the phenopacket standard encourages the use of hierarchical classification systems and vocabularies such as HPO, UBERON or the NCIT; however it does not specify which ontologies have to be used, permitting user groups exchanging data to

**Table 1.** Summary of top-level fields of a Phenopacket. With the exception of the id and the metadata, all fields are optional.

| Field | Intended use |
|---|---|
| Id | An application-specific identifier for the phenopacket |
| subject | Demographic information about the subject of the phenopacket (often a patient or proband) |
| phenotypicFeatures | List of PhenotypicFeature messages, each of which provides an ontology term and additional context to describe a phenotypic feature (symptoms, signs, laboratory, or imaging findings, etc.). |
| measurements | List of Measurement messages, each of which represents the primary results of a clinical measurement such as a laboratory test. In many cases, Measurement messages are used to store the numerical results of a test. |
| biosamples | List of Biosample messages, each of which represents clinical data about a biosample such as a biopsy. |
| interpretations | List of Interpretation messages, each of which represents the clinical interpretation of a genetic or genomic investigation such as whole-genome sequencing. |
| diseases | List of Disease messages, each of which represents a clinical diagnosis of a disease. |
| medicalActions | List of MedicalAction messages, each of which represents an action such as a treatment or clinical procedure taken for the clinical management of the subject of the phenopacket. |
| files | List of File messages, each of which represents a file with data or results from a genomic or analogous high-throughput investigation. This field is intended for investigations done on peripheral blood samples (corresponding to germline DNA in the case of whole-genome sequencing). Investigations performed on other samples should be represented within the corresponding Biosample message. |
| metaData | Information about ontologies and references used in the phenopacket. |

```
291  metaData:
292    created: "2021-05-14T10:35:00Z"
293    createdBy: "anonymous biocurator"
294    resources:
295    - id: "ncit"
296      name: "NCI Thesaurus"
297      url: "http://purl.obolibrary.org/obo/ncit.owl"
298      version: "21.05d"
299      namespacePrefix: "NCIT"
300      iriPrefix: "http://purl.obolibrary.org/obo/NCIT_"
301    - id: "efo"
302      name: "Experimental Factor Ontology"
303      url: "http://www.ebi.ac.uk/efo/efo.owl"
304      version: "3.34.0"
305      namespacePrefix: "EFO"
306      iriPrefix: "http://purl.obolibrary.org/obo/EFO_"
307    - id: "uberon"
308      name: "Uber-anatomy ontology"
309      url: "http://purl.obolibrary.org/obo/uberon.owl"
310      version: "2021-07-27"
311      namespacePrefix: "UBERON"
312      iriPrefix: "http://purl.obolibrary.org/obo/UBERON_"
313    - id: "ncbitaxon"
314      name: "NCBI organismal classification"
315      url: "http://purl.obolibrary.org/obo/ncbitaxon.owl"
316      version: "2021-06-10"
317      namespacePrefix: "NCBITaxon"
318      iriPrefix: "http://purl.obolibrary.org/obo/NCBITaxon_"
319    phenopacketSchemaVersion: "2.0"
```

**Figure 12.** Phenopackets Metadata. The MetaData message contains information about each ontology used to provide terms in the phenopacket as well as the phenopacket version. Version 1 of the GA4GH standard was released in 2019 to elicit feedback from the community. Version 2 was developed on the basis of this feedback and is described here.

select the most appropriate and information-rich terminologies. Although the example shown here was comprehensive, for many current analysis programs in use in human genetics, a simple list of HPO terms and a path to the VCF file is all that is required to run them. We anticipate that the GA4GH Phenopacket Schema will enable such programs to use additional information for computational variant prioritization, such as excluded features, age of onset, and measurements. The Phenopacket Schema represents numerous categories of data important for translational research and diagnostics (**Table 1** provides a summary). The Phenopacket does not represent all relevant data; two notable omissions that

may be addressed in future versions of the Phenopacket Schema are social determinants of health (SDoH) and environmental exposures.

In general, computational tools are required to create and use phenopackets. Although the Phenopacket Schema is a relatively new development, several tools are already available. PhenoTips (https://phenotips.org/) is an open-source software tool for collecting and analyzing phenotypic information from probands or families with a suspected genetic condition.[32] PhenoTips can be used to collect HPO terms and other information and can export the data as a phenopacket. SAMS (Symptom Annotation Made Simple) is a freely available web-based application that offers a number of features including the ability to create phenopackets on the fly or to export case-level data as phenopackets.[33] Additionally, SAMS enables patients or relatives to enter data to be shared with physicians and offers an API that can be integrated into other applications.

A software library called phenopacket-tools (https://github.com/phenopackets/phenopacket-tools) provides developers with resources to streamline the creation and validation of phenopackets. The Phenopacket provides a standard input format for these tools that will simplify computational analysis pipelines, especially if the steps in the pipeline include a comparison of the results of multiple tools. Exomiser,[34] LIRICAL,[29] Phen2Gene,[30] and CADA[31] can take Phenopackets as input files, and other analysis tools will soon accept phenotype data in Phenopacket format.

As clinicians move from traditional clinical annotation practices towards more routine use of computationally powerful representations of concepts in medicine this flexibility will facilitate the development and adoption of a rapidly growing ecosystem of digital support tools, and will empower forward-looking computational multimodal data discovery and analysis. Genomic data will become ever more important in translational research and clinical care in the coming years and decades. The Phenopacket Schema represents a standard for capturing clinical data and integrating it with genomic data that will help to obtain the

maximal utility of this data for understanding disease and developing precision medicine approaches to therapy.

## Supporting Information

Supporting Information is available from the Wiley Online Library or from the author.

## Acknowledgements

NIH NHGRI RM1HG010860, NIH NHGRI 1R01HG011799-01, NIH OD R24OD011883, NIH NICHD 1R01HD103805-01, NIH NLM contract #75N97019P00280. Support was provided by EU Horizon 2020 research and innovation programme grant agreement 779257 (SOLVE-RD). P.N.S. acknowledges the support of the Alan Turing Institute.

## Conflict of Interest

The authors declare no conflict of interest.

## Data Availability Statement

The data that support the findings of this study are available in the supplementary material of this article.

## Peer Review

The peer review history for this article is available in the Supporting Information for this article.

## Keywords

deep phenotyping, FAIR data, Global Alliance for Genomics and Health, Human Phenotype Ontology, Phenopacket Schema

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

**2200016 (12 of 12)**

