## [**Supplementary Information**: Record of Transparent Peer Review · Advanced Genetics]

GA4GH Phenopackets: A practical introduction

Markus S. Ladewig, Julius O.B. Jacobsen, Alex H. Wagner, Daniel Danis, Baha El Kassaby, Michael Gargano, Tudor Groza, Michael Baudis, Robin Steinhaus, Dominik Seelow, Nikolaos E. Bechrakis, Christopher J. Mungall, Paul N. Schofield, Olivier Elemento, Lindsay Smith,1 Julie A. McMurry, Monica Munoz-Torres, Melissa A. Haendel, Peter N. Robinson*

* Corresponding

Review timeline:	Date submitted:	26-Apr-2022
	1 st Editorial Decision:	18-May-2022
	Revision Received:	16-Jun-2022
	2 nd Editorial Decision:	29-Jun-2022
	Revision Received:	30-Jun-2022
	Accepted:	01-Jul-2022

Editor: Kerstin Brachhold

1 st Peer Review	26-Apr-2022 to 18-May-2022
----------------------------

Reviewer #1

Overall, this manuscript on practical introduction to GA4GH Phenopackets is very well written and clear, and the concept of phenopackets and continuous effort of the GA4GH to enable easier synchronization and communication/sharing of BIG data is enormously appreciated, thank you. The choice to use RB patient to introduce step by step Phenopackets is excellent for it's dual RD and cancer-related connections.

There are several places in the manuscript that would benefit from additional explanation/clarity:

1. Fields/elements

a. Biosample

i. Would a blood/saliva also be considered as a biosample source for detection of germline variants?

b. Disease

i. If the disease = clinical diagnosis, does this mean that Phenopacket only includes individual with clinical diagnosis? What if the individual has a phenotype where clinical diagnosis is difficult to establish?

c. MedicalAction

i. Does medical action allow for integration of evolving n=1 drugs, like ASOs for RDs?

ii. Is there a field to indicate the time of onset of adverse events?

2. Text:

a. Case report/Interpretation

- i. What is considered mosaic 13q germline deletion? How is that different from a somatic variant that occurs early during the embryogenesis, so 40% rather than 50% ratio?
- ii. Did tumour cells contain the deletion in addition to the somatic missense variant?

3. Figures

- a. Fig. 2 It is unclear what the arrows represent? Why does the "Disease" element have no arrows feeding into it?
- b. Figure 13 - in the legend lines 275-305 are described, but the image only depicts lines from 291 onward.

Reviewer #2

The article entitled « GA4GH Phenopackets: A practical introduction" present a Standard for sharing disease and phenotype information. The article is essentially a demonstration of the versatility of the standard by the use of one example. I'm a clinician with background in Mendelian disease, and I think it's a great to propose a common way to share information for rare disease. However it's unclear how a lay clinician will be able to use this packet. Indeed most write in natural language and share information in natural language sometime with very nonstandard abbreviation. The discussion should more clearly stated how a lay clinician is supposed to write a patient phenopacket, will there be proposed On-Line interface ? Moreover would the authors suggest to have a mandatory phenopackets for case report or be included in CARE guideline.

In conclusion the standard is interesting but some description of the use in practical setting could be interesting

Reviewer #3

With interest I read the manuscript by Ladewig et al, providing an introduction into the GA4GH Phenopackets structure. The manuscript is overall well-written, and may serve as a practical guide on the demonstration of how to use/catalog phenotypic information as well as many other traits in a standardized fashion. The latter makes the data compliant to the FAIR principles and excellent use in clinical settings and research applications. Whereas I without any doubt underscore the importance of the principle of phenopackets, I have some suggestions/comments/questions for the authors that require some further information.

General comments

1. The manuscript does not explicitly mention the compliance to FAIR principles as advantages of its use. With many clinical laboratories and research organisations propagating the FAIR principles, it seems a missed opportunity to not focus on these aspects for the widespread adaptation/implementation of PhenoPackets
2. The term 'PhenoPackets' does not do full justice to what the packets offer: it is (much!) more than capturing the phenotypic information, as it also catalogs disease status, and medical actions. The initial connotation limits this to phenotypic information
3. The use of a case report to explain the use of PhenoPackets is attractive - yet, it might be useful to also (in the discussion) highlight (if possible) some real live use cases. Are there any

project/programs/hospitals that have already adopted to the use of PhenoPackets to capture the information presented? And if so, what are their experiences?

More detailed remarks

1. Keywords for the paper:

2. Suggestion to remove 'retinoblastoma' to be replaced by something more general such as 'FAIR data' (as was also used for the manuscript classification)

3. Abstract:

The abstracts misses a statement on why GA4GH promotes this standard. Consider adding such a general comment as final sentence to the abstract.

4. Introduction:

Page 3: Lines 33-55 provide very detailed information on retinoblastoma, but may be better incorporated with page 4, lines 49-65 and page 5, lines 1-3 because of redundancies between both sections.

5. Overall structure of the PhenoPacket/Figure 2. Whereas I acknowledge the use of figure 2 as an example of a PhenoPacket structure, it does raise some questions:

- The legend does not provide an explanation for the purple coloured boxed (e.g.

Biosample/measurement)

- It is unclear why there is no arrow from 'Phenotypic Feature' to 'Individual' as he/she may also already have features that can be captured by e.g. HPO, that do not necessarily belong to the biosample of the individual

- The arrow from gestational age to time element makes sense. However, this time element is very much specific to the Individual rather than the biosample of an individual. Consider an additional arrow from time element to individual, or the removal of gestational age.

- Why is genomic interpretation only part of diagnosis - there may well be variants that are not part of a diagnosis? I would have expected 'genetic information' or 'genetic information' as a higher order category than where it is currently positioned.

- I understand what the authors mean by using the term 'karyotypic sex' (and this is also addressed in the section 'Individual', but with other technologies than karyotyping being used much more often, a term such as 'genetically determined sex' seems more appropriate for the figure, and the example of 'karyotypic sex' can then be explained in the section 'Individual' as pragmatic example.

6. Individual:

- The authors rightfully address the GDPR related points with, for instance, respect to date of birth. From a PhenoPackets points of view when integrated within an EHR - I hope the authors can comment on that such information can be automatically retrieved from the EHR and incorporated from this - rather than having to provide the information (by hand) again? In addition, I would assume that having the information in PhenoPackets allows the users of exporting the detailed information (i.e. exact date of birth) in modified form (ie. Birthyear only from the exact date of birth that is listed). This would enhance its use for research purposes while maintaining a rich dataset at the source.

- The individual is often a proband. For use cases within the field of rare disease, often families are used (e.g. multiple affected individuals for AD inheritance, or trios in case of sporadic disease caused by de novo mutations). Initially I wondered whether trios, or other family structure can be captured. Only

much later in the manuscript, the pedigree option is listed. I would recommend placing the pedigree section as a natural follow-up (or even within the section) Individual.

- For the time stamp in Figure 3B I wonder what this time stamp refers to. If is so accurate (at level of seconds) that I wonder whether this is not the timestamp that reflects to 'at this [timestamp] moment, I saved the information in a database that this person is deceased [i.e. EHR]', but not necessarily the biological time point the patient deceased. Consider a more practical example?

7. Phenotypic features:

- Consider replacing the clinodactyly example in Fig 4 by another phenotype than an HPO phenotype to better reflect the diversity of phenotypic features as presented in the text of the paragraph. Also, the clinodactyly is not an example of an incidental finding, as this was something known the be identifiable from an physical examination (e.g. not incidental). Also, clinodactyly occurs in ~3% of the population and may thus be considered a common trait rather than an condition. Yet, what authors likely mean is that this is not associated with the RB1 variant(s). More importantly, later on in the example, CREBBP is mentioned, and, for the associated Rubenstein-Raybi (RSTS1), clinodactyly is known as feature.

8. Measurements:

- Again a well-presented example! It however again makes me wonder on the administrative/hands-on work to capture the measurements for each individual. Is there a way to automatically insert/retrieve information for normal measurements, or is this something that has to be addressed per measurement, which would make it very time consuming, and error-prone. In addition, it the absolute value is of course of essence, yet, it in the measurement nowhere lists that the pressure is too high. One needs to deduce this information from interpreting the value (i.e. 25 for the left eye) in comparison to the normal range (ie. 10-21). For the Phenotypic features, the 'conclusions of the observations are listed' (ie. the presence/absence of a feature). Would it be possible to also include this 'conclusion' for the measurements? Or would this be then classified as a 'phenotypic feature' [i.e. too high ocular pressure']. Now, the retinal detachment (possibly due to the ocular hypertension) is listed. I understand that a PhenoPacket itself does not 'draw conclusions' or finds relations between features/measures, it made me wonder how to classify this information and where it would belong. That is, PhenoPacket should/would enhance standardization, but no such information may be linked to different high level themes, and I wonder what those consequences would be.

9. Interpretation

- For the example in Figure 8 is not clear on which BioSample from patient A the data was derived. Consider the addition of the BioSample.

- What is the rational to call the variant in Figure 9 a heterozygous variant, as it was said to be mosaic in the text? A mosaic variant occurs on one allele, but is not heterozygous for its zygotity state.

10. Figure 12: The legend refers to the fieldattributes field, but this is not visible in the figure (but the fileattributes is).

1 st Editorial Decision	18-May-2022
Editorial Decision: Revise and resubmit after addressing the reviewers' comments	
Recommendation of the reviewers	
Reviewer #1 Recommends Minor Revision	
Reviewer #2 Recommends Minor Revision	
Reviewer #3 Recommends Minor Revision	

Reviewer #1:

Overall, this manuscript on practical introduction to GA4GH Phenopackets is very well written and clear, and the concept of phenopackets and continuous effort of the GA4GH to enable easier synchronization and communication/sharing of BIG data is enormously appreciated, thank you. The choice to use RB patient to introduce step by step Phenopackets is excellent for it's dual RD and cancer-related connections.

Response: Thank you for the positive assessment.

Biosample: Would a blood/saliva also be considered as a biosample source for detection of germline variants?

Response:

Any source of germline DNA can be used. If the source is considered irrelevant, it does not need to be further specified (For instance, most publications in human genetics do not specify that germline DNA was obtained from a peripheral blood sample and that the blood sample was collected in an EDTA tube etc etc). If this is required, then the Biosample element can be used.

We have added an explanation in the **'Biosample' section:**

While a Biosample message could be used to describe routine blood or buccal swab samples used for analysis of germline DNA, if the specifics of the blood or buccal swab sample are not deemed relevant for the analysis, the information can be omitted (in this case, the `subjectOrBiosampleId` of the relevant `GenomicInterpretation` element is set to the proband id; see below).

Disease: If the disease = clinical diagnosis, does this mean that Phenopacket only includes individuals with clinical diagnosis? What if the individual has a phenotype where clinical diagnosis is difficult to establish?

Response:

This element is not required and can be omitted if the diagnosis is unknown. It is also possible to put a general, working diagnosis if the molecular diagnosis is unknown. We have added text about this to the section on the *Disease* element:

In cases where there is no disease diagnosis, the Disease message can be omitted.

Alternatively, a general diagnosis can be indicated by choosing an ontology term that denotes a group of specific diseases. For instance, the initial clinical diagnosis of *ciliopathy* (MONDO:0005308) might be made in a child with clinical manifestations such as Situs inversus and postaxial polydactyly. The genetic workup (which would be represented in the *GenomicInterpretation* message; see below) could describe the molecular diagnosis of *Bardet-Biedl syndrome 17* (MONDO:0014445).

MedicalAction

- i. Does medical action allow for integration of evolving n=1 drugs, like ASOs for RDs?
- ii. Is there a field to indicate the time of onset of adverse events?

Response:

This element can be used for any treatment for which an Ontology term can be found (or created). The Phenopacket Schema does not constrain users to a specific terminology. In practice, most users will use established terminologies such as DrugBank or DrugCentral, which may limit available terms to relatively well established medications.

There is no field to indicate the time of onset of adverse events. If this turns out to be a general need, it could be added to a future version of the Schema.

What is considered mosaic 13q germline deletion? How is that different from a somatic variant that occurs early during the embryogenesis, so 40% rather than 50% ratio?

Response:

This is in fact a somatic variant that (must have) occurred early in embryogenesis so that not all cells of the body carry the deletion. We note that we are constructing this example from several published cases. However, it is the case in retinoblastoma that the tumor cell contains both the germline mutation as well as the somatic (acquired) mutation.

We added a brief explanation about somatic mosaicism to the text (end of the introduction):

Mosaicism results from postzygotic mutation that occurs during early embryonic development and can lead to germline or somatic mosaicism, potentially causing a less severe and/or variable phenotype compared with the equivalent constitutive mutation^{[12]}.

Did tumour cells contain the deletion in addition to the somatic missense variant?

Response:

We note that we are constructing this example from several published cases. However, it is the case in retinoblastoma that the tumor cell contains both the germline mutation as well as the somatic (acquired) mutation.

Fig. 2 It is unclear what the arrows represent? Why does the "Disease" element have no arrows feeding into it?

Response:

The *Disease* element contains simple fields such as strings (i.e., labels) or ontology terms. Many of the other elements are more complex. For instance, the *Biosample* can contain *PhenotypicFeature* and *Measurement* elements and for this reason has arrows.

Figure 13 - in the legend lines 275-305 are described, but the image only depicts lines from 291 onward.

Response:

Thank you for pointing out this error. We have fixed the legend text.

Reviewer #2:

The article entitled « GA4GH Phenopackets: A practical introduction" present a Standard for sharing disease and phenotype information. The article is essentially a demonstration of the versatility of the standard by the use of one example. I'm a clinician with background in Mendelian disease, and I think it's a great to propose a common way to share information for rare disease. However it's unclear how a lay clinician will be able to use this packet. Indeed most write in natural language and share information in natural language sometime with very nonstandard abbreviation. The discussion should more clearly stated how a lay clinician is supposed to write a patient phenopacket, will there be proposed On-Line interface ? Moreover would the authors suggest to have a mandatory phenopackets for case report or be included in CARE guideline.

In conclusion the standard is interesting but some description of the use in practical setting could be interesting

Response:

Thank you for this assessment. The Phenopacket was introduced very recently, but there are already a number of software tools that can be used to create phenopackets. Our intention with this article is to provide clinicians, researchers, and software developers with an explanation of the contents of a phenopacket rather than to explain how to create them, which will be different with each tool.

We thank you for noticing that the article did not state this clearly, and we have added an additional paragraph about this to the discussion:

In general, computational tools are required to create and use phenopackets. Although the Phenopacket Schema is a relatively new development, several tools are already available.

PhenoTips (<https://phenotips.org/>) is an open-source software tool for collecting and analyzing phenotypic information from probands or families with a suspected genetic condition^[32].

PhenoTips can be used to collect HPO terms and other information and can export the data as a phenopacket. SAMS (Symptom Annotation Made Simple) is a freely available web-based application that offers a number of features including the ability to create phenopackets on the fly or to export case-level data as phenopackets^[33]. Additionally, SAMS enables patients or relatives to enter data to be shared with physicians and offers an API that can be integrated into other applications.

A software library called phenopacket-tools (<https://github.com/phenopackets/phenopacket-tools>) provides developers with resources to streamline the creation and validation of phenopackets. The Phenopacket provides a standard input format for these tools that will simplify computational analysis pipelines, especially if the steps in the pipeline include a comparison of the results of multiple tools. Exomiser^[34], LIRICAL^[35], Phen2Gene^[30], and CADA^[31] can take Phenopackets as input files, and other analysis tools will soon accept phenotype data in Phenopacket format.

Reviewer #3:

With interest I read the manuscript by Ladewig et al, providing an introduction into the GA4GH Phenopackets structure. The manuscript is overall well-written, and may serve as a practical guide on the demonstration of how to use/catalog phenotypic information as well as many other traits in a standardized fashion. The latter makes the data compliant to the FAIR principles and excellent use in clinical settings and research applications. Whereas I without any doubt underscore the importance of the principle of phenopackets, I have some suggestions/comments/questions for the authors that require some further information.

Response:

Thank you for your suggestions!

General comments

1. The manuscript does not explicitly mention the compliance to FAIR principles as advantages of its use. With many clinical laboratories and research organisations propagating the FAIR principles, it seems a missed opportunity to not focus on these aspects for the widespread adaptation/implementation of PhenoPackets

Response:

We agree. We have added a short reference to FAIR at the top of the introduction.

The Phenopacket schema supports the FAIR principles (Findable, Accessible, Interoperable, and Reusable), and computability.^[2-5] Specifically, Phenopackets are designed to be both human and machine-interpretable, enabling computing operations and validation on the basis of defined relationships between diagnoses, lab measurements, and genotypic information.

2. The term 'PhenoPackets' does not do full justice to what the packets offer: it is (much!) more than capturing the phenotypic information, as it also catalogs disease status, and medical actions. The initial connotation limits this to phenotypic information

Response:

This is a fair point. Indeed the phenopacket intends to represent more than phenotypic features. However, the format has grown over the last seven years and it is already in wide use, so it was no longer possible to change the name of the standard!

3. The use of a case report to explain the use of PhenoPackets is attractive - yet , it might be useful to also (in the discussion) highlight (if possible) some real live use cases. Are there any project/programs/hospitals that have already adopted to the use of PhenoPackets to capture the information presented? And if so, what are their experiences?

Response:

We have added a paragraph about current software tools that have adopted the Phenopacket Schema (see our response to Reviewer #2, above).

1. Keywords for the paper: Suggestion to remove 'retinoblastoma' to be replaced by something more general such as 'FAIR data' (as was also used for the manuscript classification)

Response:

Agree, thank you for the suggestion!

3. The abstracts misses a statement on why GA4GH promotes this standard. Consider adding such a general comment as final sentence to the abstract.

Response:

Thank you for the suggestion. We have added this final sentence to the abstract:

The Phenopacket Schema, together with other GA4GH data and technical standards, will enable data exchange and provide a computational foundation for the analysis of disease and phenotype information to improve our ability to diagnose and conduct research on all types of diseases, including cancer and rare disease.

Page 3: Lines 33-55 provide very detailed information on retinoblastoma, but may be better incorporated with page 4, lines 49-65 and page 5, lines 1-3 because of redundancies between both sections.

Response:

The purpose of providing this information as a quotation was to give the reader an idea of the raw data used to create the phenopacket. To improve the flow of the manuscript, we moved the description of the case report beneath the general introduction to the start of the description of the individual elements.

5. Overall structure of the PhenoPacket/Figure 2. Whereas I acknowledge the use of figure 2 as an example of a PhenoPacket structure, it does raise some questions:

- The legend does not provide an explanation for the purple coloured boxed (e.g. Biosample/measurement)

Response:

This was an oversight on our part. We have updated the legend of the Figure.

- It is unclear why there is no arrow from 'Phenotypic Feature' to 'Individual' as he/she may also already have features that can be captured by e.g. HPO, that do not necessarily belong to the biosample of the individual

Response:

The Individual message contains demographic information about the patient or person represented by the Phenopacket and is not intended to represent all the information about the individual. To clarify this, we have added the following sentence to the section in Individual.

All information in a phenopacket refers to this individual.

- The arrow from gestational age to time element makes sense. However, this time element is very much specific to the Individual rather than the biosample of an individual. Consider an additional arrow from time element to individual, or the removal of gestational age.

Response:

The TimeElement can be used with almost all of the other major messages of the Phenopacket Schema, and the Figure is a simplified representation. The TimeElement is always assumed to represent the age of the individual at the time of some observation, so even if say we attach a TimeElement to a Biospecimen, we are not referring to the “age” of a biopsy but to the Age of the patient at the time the biopsy was taken.

- Why is genomic interpretation only part of diagnosis - there may well be variants that are not part of a diagnosis? I would have expected 'genetic information' or 'genetic information' as a higher order category than where it is currently positioned.

Response:

The Phenopacket Schema makes choices about what to include does not provide a comprehensive model of genetic information, instead it focuses on genetic interpretations deemed relevant to a clinical diagnosis. If desired, Phenopackets could be used as a part of a larger data ecosystem with more genetic information, but that is beyond the scope of the current manuscript.

I understand what the authors mean by using the term 'karyotypic sex' (and this is also addressed in the section 'Individual', but with other technologies than karyotyping being used much more often, a term such as 'genetically determined sex' seems more appropriate for the figure, and the example of 'karyotypic sex' can then be explained in the section 'Individual' as pragmatic example.

Response:

Genetically determined sex is not the same as karyotypic sex. For instance, Disorders of sex development (DSD) associated with variants in *SRY* or *SOX9*. The karyotypic sex field was requested by participants in the GA4GH workshops for phenopackets to represent the result of chromosome analysis (karyotyping) of the sex chromosomes. If appropriate, there are HPO terms that represent many DSD findings.

Phenotypic features: Consider replacing the clinodactyly example in Fig 4 by another phenotype than an HPO phenotype to better reflect the diversity of phenotypic features as presented in the text of the paragraph. Also, the clinodactyly is not an example of an incidental finding, as this was something known to be identifiable from a physical examination (e.g. not incidental). Also, clinodactyly occurs in ~3% of the population and may thus be considered a common trait rather than a condition. Yet, what authors likely mean is that this is not associated with the RB1 variant(s). More importantly, later on in the example, CREBBP is mentioned, and, for the associated Rubenstein-Raybi (RSTS1), clinodactyly is known as a feature.

Response:

We write “Clinodactyly, which is not known to be related to retinoblastoma and is presumably an incidental finding, was noted at the age of 3 months (P3M).” -- As the reviewer states, we used “incidental” to mean “likely occurring by chance” meaning that it is not a consequence of retinoblastoma, and we think this is an idiomatic use of the word. In general, we recommend that all phenotypic features found on examination be included in phenopackets, and not only those that the physician deems to be relevant -- otherwise, biases will ensue that could make it difficult to discover novel associations. In this particular case, Clinodactyly was noted in the original case report we are modeling and so we would prefer to keep this term.

Measurements:

- Again a well-presented example! It however again makes me wonder on the administrative/hands-on work to capture the measurements for each individual. Is there a way to automatically insert/retrieve information for normal measurements, or is this something that has to be addressed per measurement, which would make it very time consuming, and error-prone. In addition, if the absolute value is of course of essence, yet, it in the measurement nowhere lists that the pressure is too high. One needs to deduce this information from interpreting the value (i.e. 25 for the left eye) in comparison to the normal range (ie. 10-21). For the Phenotypic features, the 'conclusions of the observations are listed' (ie. the presence/absence of a feature). Would it be possible to also include this 'conclusion' for the measurements? Or would this be then classified as a 'phenotypic feature' [i.e. too high ocular pressure]. Now, the retinal detachment (possibly due to the ocular hypertension) is listed. I understand that a PhenoPacket itself does not 'draw conclusions' or finds relations between features/measures, it made me wonder how to classify this information and where it would belong. That is, PhenoPacket should/would enhance standardization, but no such information may be linked to different high level themes, and I wonder what those consequences would be.

Response:

Indeed, in many cases, *Measurements* could be directly extracted from existing data (e.g., electronic healthcare records). We envision that many software tools will be developed for creating phenopackets in various settings and have now added a description of several existing tools that can be used to create phenopackets or work with them (See our response to Reviewer #2).

It would also be possible to computationally infer HPO terms based on abnormal Measurements. For instance, if the intraocular pressure is abnormally low, we could infer *Low intraocular pressure* (HP:0032547). There are situations where it may be important to retain the original data and those where it is preferable to use ontology terms. This gets into computational

analysis and algorithms, which is beyond the scope of this paper but certain was actually the motivation for developing the schema!

Interpretation: For the example in Figure 8 is not clear on which BioSample from patient A the data was derived. Consider the addition of the BioSample.

Response:

The field "subjectOrBiosampleId" is used to denote the source of the genomic interpretation. In Figure 9, it is shown as "biosample.1" which corresponds to the Biosample of Figure 6.

In the legend to Figure 6 we had this: The interpretation based on this sequencing is presented in Figure 9.

We have revised the legend to Figure 9 as follows for clarity:

The GenomicInterpretation message uses the subjectOrBiosample field to indicate the source of the sequenced material. In this case, the source was the tumor specimen described in the Biosample message of Figure 6.

What is the rationale to call the variant in Figure 9 a heterozygous variant, as it was said to be mosaic in the text? A mosaic variant occurs on one allele, but is not heterozygous for its zygosity state.

Response:

Mosaicism occurs at the level of cells, and it is not uncommon to see things like this "The quantification was confirmed on 7 independent replications, consistent with cells mosaic for heterozygous and homozygous STXBP1 mutations in the dysplastic tissue." Basically, any one cell either has the mutation or not. If the mosaicism involves half of the cells, then a heterozygous mutation will have an allele frequency of 25%. In practice we do not observe whether 25% of cells have a homozygous mutation or 50% of cells have a heterozygous mutation, but in the current case the interpretation infers the variant is heterozygous because this is characteristic of retinoblastoma. (Any retinal cell with a homozygous mutation would go on to develop into a tumor, and so a mosaicism for a homozygous mutation would lead to multifocal tumors).

10. Figure 12: The legend refers to the field attributes field, but this is not visible in the figure (but the fileattributes is).

Response:

Thank you for picking up our error. It should have been "fileAttributes" in the legend. We have corrected it.

Reviewer #1

The revised manuscript addresses well reviewers comments, concerns and suggestions. I only have a few minor comments:

Referencing: The edited text and corresponding citations do not seem to follow overall style (e.g. p.3 Line 17 [2-5] after the period and not in superscript. Please review carefully for consistency.

Writing: The manuscript needs proofreading (e.g. p.7 line 19 double "is"; p.8 starting line 42 some text is bolded, and it is unclear why)

Reviewer #2

The author answer clearly my question

Reviewer #3

The authors have addressed all the queries/comments sufficiently. I have no further comments.

Second Editorial Decision	29-Jun-2022
Editorial Decision: Revise and resubmit after addressing the final minor comments of reviewer #1	
Recommendation of the reviewers	
Reviewer #1 Recommends Minor Revision	
Reviewer #2 Recommends Acceptance	
Reviewer #3 Recommends Acceptance	

Authors' Response to 2nd Review	16-Jun-2022
-------------

Minor errors corrected as suggested

Final Decision	01-Jul-2022
----------------	-------------

Accept the revised version for publication as the authors satisfactorily addressed the final minor comments of reviewer #1.